# Adapting Overwintering Honey Bee (*Apis mellifera* L.) Colony Management in Response to Warmer Fall Temperatures Associated with Climate Change

**DOI:** 10.3390/insects16030266

**Published:** 2025-03-04

**Authors:** Gloria DeGrandi-Hoffman, Henry Graham, Vanessa Corby-Harris, Mona Chambers, Emily Watkins-deJong, Kate Ihle, Lanie Bilodeau

**Affiliations:** 1Carl Hayden Bee Research Center, USDA-ARS, 2000 East Allen Road, Tucson, AZ 85719, USA; ricgra10@gmail.com (H.G.); vanessa.corbyharris@usda.gov (V.C.-H.); mona.chambers@usda.gov (M.C.); emily.watkinsdejong@usda.gov (E.W.-d.); 2Honey Bee Breeding, Genetics, and Physiology Research Lab, USDA-ARS, 1157 Ben Hur Rd., Baton Rouge, LA 70820, USA; kate.ihle@usda.gov (K.I.); lanie.bilodeau@usda.gov (L.B.)

**Keywords:** overwintering, climate change, Varroa, cold storage, fat body, almond pollination, beekeeping costs

## Abstract

Climate change is causing warmer fall temperatures that can lead to overwintering losses of honey bee colonies. Warmer temperatures cause honey bees to fly and parasitic Varroa mites to immigrate into colonies later in the fall. A management strategy that might reduce colony losses is the combination of Varroa-resistant Russian honey bees and overwintering colonies in indoor cold storage facilities. Using this strategy, we found that Russian bees overwintered in cold storage have similar survival, colony sizes, and percentages that could be rented for almond pollination to unselected European colonies. An analysis of fat body metrics, which are key to overwintering survival, showed similar trends between Russian and unselected bees. A cost comparison of overwintering Russian bees in cold storage versus apiaries showed that overwintering in cold storage costs less than in apiaries. The percentages of colonies that survived and were rented for almond pollination were similar between overwintering in cold storage or apiaries. The combination of Varroa-resistant Russian bees and cold storage overwintering can be a viable management strategy for mitigating the effects of climate change on colony survival.

## 1. Introduction

Climate change impacts all sectors of society including agriculture and food security. How and where crops are grown now, and in the future, depends on adapting cultural practices to an environment where temperature, rainfall, and the frequency of extreme weather events are rapidly changing. The beekeeping industry is not immune to the effects of climate change. Colony losses have remained at unsustainable levels since 2006 when data were first recorded [1]. These losses reverberate across agriculture because honey bees pollinate more than one-third of all crops that support a range of economic sectors of considerable monetary value [2]. Human health is also affected because insufficient pollination reduces yields and the availability of fruits, oilseeds, and vegetables that are rich in nutrients needed to maintain cardiovascular health and prevent cancer and metabolic diseases [3].

Though colonies are lost throughout the year due to poor nutrition, pests, pathogens, and pesticides, most losses occur over the winter [4,5]. Fall temperatures, particularly in temperate areas, are trending higher than historical averages and are predicted to follow this course into the future [6,7]. Warmer fall temperatures can threaten colony survival in at least two ways. First, warm temperatures stimulate foraging and leave colonies vulnerable to immigration by parasitic Varroa mites (*Varroa destructor* Anderson and Trueman) [8,9]. Varroa is an external parasite of honey bees that parasitizes all stages of workers and drones, reproduces while feeding on pupae under sealed comb cells, transmits viruses, and is the primary cause of overwintering colony losses [10,11,12]. Varroa migrate into colonies by attaching to foraging bees. Though some migration might happen throughout the spring and summer, the greatest frequency of foragers carrying mites into colonies occurs in the fall and is correlated with sharp increases in mite infestations [13]. Since mites can immigrate into colonies whenever foraging weather occurs, warm fall temperatures late into the season extend the period of mite migration. Warmer temperatures might also prolong brood rearing and mite reproduction, further adding to the difficulties in keeping Varroa at low levels prior to overwintering.

A second way that warmer fall temperatures in temperate areas can threaten colony survival is their effects on the age structure of overwintering colony populations [7,14]. In the fall, colonies in temperate areas prepare for winter by storing resources and slowing the production of new bees (i.e., brood rearing) [15]. If temperatures are below 10 °C, the bees remain in the hive and form a thermoregulated cluster. However, warm temperatures cause bees to fly, and flight activity physiologically ages bees [16,17,18,19]. Greater flight activity in the fall can cause overwintering populations to contain higher proportions of bees that are physiologically older. When colonies resume foraging in the spring, older bees can die at rates that exceed the colony’s ability to replace them, so the colony declines. Colony decline after overwintering is common and is part of the yearly population dynamics of a colony [14]. However, extended periods of fall foraging can lead to a deep population decline in the spring from which the colony cannot recover. Indeed, years with higher colony losses are correlated with warmer and drier winter conditions [20,21].

The effects that climate change is having on Varroa control might be mitigated with genetic lines of Varroa-resistant or -tolerant honey bees. Beekeepers currently use Varroa-tolerant or -resistant bees to reduce costs and prevent Varroa resistance to miticide treatments that can be needed at frequent intervals from spring to fall. One Varroa-resistant line is Russian bees brought into the U.S. in 1997 [22]. The original queens were from regions in Russia where bees developed resistance to Varroa through natural selection [23]. Varroa populations are kept low because brood in Russian colonies is unattractive to mites, so reproduction is suppressed [24,25]. Russian bees also remove larvae from infested brood cells (hygienic behavior) and from the bodies of nestmates (grooming behavior) [23]. However, the benefits of suppressing mite reproduction could be eliminated if flight activity extends into the fall, when mite migration becomes more frequent. Varroa migration can occur in colonies of Russian bees and causes mite populations to be comparable to unselected genetic lines [26].

Late season flight and mite migration into colonies can be curtailed by overwintering hives in cold storage facilities. Cold storage differs from overwintering in outdoor apiaries in that hives are moved into refrigerated buildings that are kept dark and maintained at <7.2 °C. Bees remain in a cluster inside the hive while in cold storage. In contrast, colonies overwintered outdoors can experience variable temperatures and fly when weather conditions are suitable. From late fall to early spring, supplemental feeding may be needed if there is no forage available. Mite numbers may rise due to mite migration if temperatures are suitable for flight, and varroacide applications might be required.

The number of hives overwintered in cold storage has increased in the U.S., because it can improve overwintering survival and be less expensive than wintering colonies in outdoor apiaries in the southern U.S. [27,28]. However, little is known about how well Russian bees will perform during cold storage. Russian bees currently are overwintered in outdoor apiaries, particularly in the southern U.S., where floral resources can be available and brood can be reared for most of the winter.

A concern about putting Russian bees in cold storage is that unlike European bees commonly used in commercial beekeeping, brood rearing slows or completely stops in Russian colonies when there is no nectar flow [29]. If brood rearing does not occur, unless nectar is being collected, the colony population can plummet when foraging resumes in the spring. To prevent colonies from becoming too small to recover from losses of overwintered bees, brood rearing needs to occur while bees are still confined in the hive. European colonies in cold storage can start rearing brood in mid- to late January [30]. If Russian bees do not rear brood in cold storage, colonies will decline and be smaller with fewer foragers than European colonies during almond pollination [31].

In this study, we explored two factors that could influence the decision to overwinter Russian bees in cold storage. The first was comparing colony survival after overwintering and almond bloom and the percentage of colonies that could be rented for almond pollination between Russian bees and European genetic lines previously used in cold storage overwintering experiments [30]. We chose to extend the study beyond overwintering and into almond pollination because more than one million colonies are moved into almond orchards in California for pollination each year. We also examined fat body metrics in unselected and Russian bees before and after cold storage, as they provide evidence for the shift from summer to winter bees that is essential for successful overwintering in cold storage [30,32,33]. The second factor in this study was comparing costs and colony survival and size between Russian colonies overwintered in either cold storage or outdoor apiaries in the southern U.S. If Russian colonies could be successfully overwintered in cold storage at costs lower than or comparable to outdoor apiaries, it would provide beekeepers with management options that might mitigate the effects of climate change on colony losses.

## 2. Materials and Methods

### 2.1. Overview of the Study

Colonies were headed by either queens of European genetic lines (i.e., queen lines) open-mated from Integri Bees/Harvest Honey, INC., Danbury, TX, USA (29.212312, −95.343355), or Russian queen lines from a certified Russian queen breeder (Coy Bee Company, Wiggins, MS, USA). European queen lines (hereafter referred to as unselected bees (UNSEL)) were sourced from three apiaries in Baldwin, ND, USA (47.070751, −100.796117). Baldwin is in central North Dakota and has a continental climate, with average temperatures between 8 °C (average low) and 22 °C (average high) at the start of the study in September and 0.5 °C and 13 °C in October. Colonies headed by Russian queen lines (RUS) were sourced from two apiaries in Hebron, ND (46.867079, −101.937705). Hebron is 125 km southwest of Baldwin and has a similar climate. Average September temperatures are between 6° and 22 °C, and average October temperatures between 13° and −1 °C.

Between 31 August and 1 September, RUS colonies from two apiary sites (n = 299) were measured for colony size (combs of bees and brood) and mites per 100 bees. UNSEL colonies (n = 158 colonies) were measured and sampled for mites from 5 to 7 September. Of these, 80 colonies of either UNSEL or RUS were chosen for overwintering in cold storage because they fit our criteria of >10 combs with adult bees, and <1 mite per 100 bees. Smaller colonies or those with higher mite numbers were not used in the study as these factors affect colony size and survival during and after cold storage [11,27]. A second group of 80 colonies headed by RUS that fit the colony size and mites per 100 bee criteria were selected to overwinter outdoors in apiaries in Wiggins, MS, USA (30.726688, −88.991526) (RUSms) (Figure 1). Wiggins is located 59.5 km from the Gulf of Mexico and has a sub-tropical climate. The average low and high temperatures in November are 7° and 22 °C, in December are 4° and 18 °C, and in January are 2° and 16 °C. Forage available to bees included *Solidago* spp., *Aster* spp., and *Baccharis halmifolia* in October and November, and *Alnus serrulata*, *Acer rubrum*, *Lamium spp*., and *Ulmus* spp. in January. Brood rearing can occur throughout the winter.

Following initial measurements in September, RUS and UNSEL were measured again during the first week of October prior to moving hives into a refrigerated cold storage facility for overwintering (Bee Storages of Idaho LLC., Filer, ID, USA) (pre-cold storage measurement). Colonies were sampled for mites per 100 bees and *Nosema* spp. levels at the time of the measurements as these parasites can affect overwintering survival. While in the facilities, colonies were kept in darkness at an average temperature of 5.6 °C maintained by refrigeration. Air exchange between the facility and the outdoors was used to keep CO_2_ levels to <10,000 ppm, and relative humidity <50% due to the dry air in Idaho. Prior to being moved to cold storage, RUS and UNSEL colonies were fed sugar syrup and pollen patties (UNSEL—Global 4% pollen patties (Global Patties, Airdrie, AB, Canada); RUS—Nutra-Bee (Nutra-Bee, St. Catharines, ON, Canada)) and treated with amitraz to control Varroa. Samples of bees for the analysis of fat body metrics were taken from the center combs in the hive during the pre-cold storage measurements (hereafter referred to as pre-cold storage samples). Due to the time and expense of analyzing fat body samples, a subsample of 30 colonies per queen line was taken. A second set of bee samples was taken from RUS and UNSEL (n = 80 colonies of each) to estimate *Nosema* spp. spores. Bee samples for Nosema analyses were taken from honey frames. Colony measurements and sampling for fat body analysis were repeated using the same colonies as the pre-cold storage samples after hives were removed from cold storage and placed in almond orchards for pollination during the last week of January (post-cold storage samples). Pre- and post-cold storage samples were packed in dry ice, shipped to the USDA-ARS, Tucson, AZ, USA, and stored at −80 °C until analysis. A final colony measurement was taken after almond bloom.

### 2.2. Estimating Colony Size

Combs of adult bees and sealed and unsealed brood were measured on Langstroth deep frames (comb dimensions: 48.3 × 2.7 × 23.2 cm). Bees and brood (open and sealed) were measured in 80 colonies each of RUS and UNSEL using methods adapted from [34]. Combs were divided into one-tenth sections, and the number covered with bees or brood on both sides of a comb was summed. Measurements from all combs were totaled to estimate combs of bees and brood in a colony. During pre-cold storage measurement in October, colonies comprised two deep Langstroth hive boxes. Temperatures were below 10 °C, so the one-tenth of the comb area method for comb evaluations was not used. Instead, adult populations were measured with the California almond pollination method [35], where 75% of a Langstroth deep comb covered in adult bees is counted as a comb of bees. Brood data are not included in the October measurements as the colonies had stopped rearing brood. For estimates of adult bee populations, the top hive body was tilted forward on the pallet and frames with at least 75% bee coverage on the top and bottom of the comb were counted as a comb of bees. In the lower hive body, adult populations were estimated from adult bees covering the top of the combs. When the cluster was small and not visible from the bottom of the lower hive body and the top of the upper hive body, the two hive bodies were split apart and combs were removed to provide an accurate count. Combs with bees were totaled for the colony as an estimate of colony size. Hereafter, measurements of adult populations made using the California method are referred to as ‘combs with bees’. Combs of bees and brood were estimated after overwintering when hives were moved to almond orchards and again after almond bloom using the one-tenth of the comb area count method described above. Areas with sealed brood after cold storage were also recorded as they can be used to approximate when brood rearing began.

### 2.3. Estimating Mite Populations

Samples of adult workers were taken between 31 August and 7 September 2022 to estimate Varroa infestations. The measurements were used to select colonies for inclusion in the study. UNSEL colonies had been treated with amitraz formulations by the beekeeper throughout the summer and late September. RUS colonies were treated with amitraz formulations only in late September. Between 2 and 6 October 2022, UNSEL and RUS colonies selected to be in the study were sampled again for mites. RUS and UNSEL hives were transported and placed into cold storage for overwintering during the second week of October. During both sampling times, approximately 300 workers per colony were brushed from brood comb into jars containing 50 mL of 70% ethanol [36]. Samples were refrigerated until analyzed. Mites were counted by vigorously shaking the sample jars, and pouring the bees and alcohol into a strainer positioned over a pan. The mites that went through the strainer and into the pan were counted. Bees in the strainer were also examined for mites. All the bees in the sample were counted to estimate mites per 100 bees [36].

### 2.4. Estimating Nosema Spores per Bee

Nosema samples were processed using methods described in [30]. Briefly, 20 bees per RUS and UNSEL colony were placed in a 50 mL test tube with 20 mL of ultra-pure water and homogenized with a Brinkmann Polytron Kinematica Generator PT-DA 2120/2 WEC (Kinematica AG 6102 Malters Switzerland) for 5 s. The homogenizer was rinsed twice using a 500 mL flask with distilled water. Samples were allowed to equilibrate for 30–60 s. The supernatant was carefully removed from the center (clear area) of the sample in the test tube and placed into a 1.5 mL Eppendorf tube. The sample was kept at −20 °C until Nosema spores were estimated.

Spores per colony were estimated by transferring a 15 µL sample of supernatant to a Bright-Line hemocytometer (Hausser Scientific, Horsham, PA, USA). Spores were counted using a compound microscope (Leica CME, Buffalo, NY, USA) at 400× with phase-contrast lighting. Nosema spores were counted in 16 small squares within 5 larger squares. The final spore count was calculated by multiplying the hemocytometer counts by 50,000 [37].

### 2.5. Fat Body Analysis

Fat body metrics were measured using protocols described in [30]. Briefly, 10 worker bees from RUS and UNSEL colonies (n = 30 colonies per queen line) were placed onto a block of dry ice, and their abdomens were removed and placed ventral side up into a waxed Petri dish. The entire gut was removed, and the remaining abdominal carcass with the fat body attached was rinsed to remove the remaining gut contents and blotted dry. Ten abdomens and three chrome beads (3.2 mm) were placed in a pre-weighed reinforced polypropylene 2 mL vial (XXTuff Microvials, Bio Spec Products catalog number 330TX, Bio Spec Products, Bartlesville, OK, USA). Samples were placed into a −20 °C freezer until further processing.

Fat body weight was estimated by drying the samples in a Fisher Scientific Iso Temp oven (Thermo Fisher Scientific, Waltham, MA, USA) at 60 °C (four days). Samples were removed, and a new previously weighed lid for the vial was immediately applied. The preliminary weight of the vial with beads was subtracted from the dry weight of the vial, beads, and dried abdomens to determine the dry weight of 10 abdominal carcasses. Samples were placed into a −20 °C freezer until they were processed for protein and lipid concentrations.

Prior to protein and lipid analyses, 1 mL of PBS was added to each sample vial. The sample was homogenized with a Bio Spec Mini-Beadbeater 96 (Bio Spec Products, Distributor Cole Parmer, Vernon Hills, IL, USA) for 60 s. Samples were centrifuged (Eppendorf Centrifuge 5424, Hamburg, Germany) at 15,000 rpm for 6 min. Without disturbing the pellet, 100 µL of the supernatant was removed from the vial, just below the top layer of lipids, and placed into a vial with 900 µL of PBS with 0.8% Halt EDTA-free Protease Inhibitor Cocktail (#78437, Thermo Scientific, Rockford, IL, USA). The sample was frozen until it was processed for protein. The remaining sample was frozen until lipid analysis.

Fat body protein was analyzed with a BCA Protein Assay kit (Thermo Scientific product 23225, Waltham, MA, USA) using methods described in [30] and manufacturer instructions. Samples were analyzed in triplicate and read in a BioTek Synergy HT Microplate reader (BioTek Instruments, Winooski, VT, USA) at a wavelength of 562 nm. The absorbance values of blank wells were subtracted from each standard and sample. A second-order polynomial equation was derived from the standards and used to estimate protein concentration in the fat body samples (µg/µL). Fat bodies had protein concentrations that exceeded the range of the equation derived from the standards, so the fat bodies were plated at a 3% solution. Protein in fat body samples was estimated using the absorbance and standard equation and then corrected for the sample dilution made prior to analysis.

Lipid analysis was conducted using methods described in [30]. Sample vials were vortexed and poured into Fisher Brand 10 mL disposable 13 × 100 mm culture vials (Thermo Fisher Scientific, Waltham, MA, USA). The sample was vortexed and a 2:1 mixture of chloroform–methanol (1 mL) was added to the culture vial along with 210 µL of 0.25% KCI. The mixture was vortexed and centrifuged at 2000 rpm for 15 min (Thermo Scientific Sorvall ST 16 centrifuge, Thermo Fisher, Waltham, MA, USA). The bottom chloroform layer was removed and placed into a new a 2 mL glass screw cap vial (Sun Sri Part # 500 306, Sun SRI, Rockwood, TN, USA), and the process was repeated. Serial dilutions of corn oil dissolved in chloroform were prepared (300, 150, 75, 37.5, 18.75, 9.4, 4.7, and 0 µg/µL) to construct a standard equation for lipid concentrations. The negative control consisted of 100 µL of chloroform. Samples, standards, and the negative control were dried to completion for approximately 1.5 h (Savant SPD 2010 Speedvac Concentrator, Thermo Fisher Scientific, Waltham, MA, USA). Dried samples were reacted with 182 µL of concentrated sulfuric acid at 100 °C for 15 min (Heating block Model HP30A, Torrey Pines Scientific, Carlsbad, CA, USA) and 1478 µL of vanillin–phosphoric acid for 15 min in the dark at room temperature. Each of the negative controls, standards, and samples (100 µL) was plated in triplicate and read using a spectrophotometer at 525 nm (Agilent BioTek Synergy HT, Santa Clara, CA, USA). The average absorbance value for the negative control was subtracted from each standard and sample. A linear equation of the standards was derived to infer the µg/µL of lipid per sample.

### 2.6. Statistical Analysis

All statistical analyses were conducted using Minitab-22.1.0 (Minitab, LLC, State College, PA, USA) and JMP-18.0 (SAS Institute, Cary, NC, USA) statistical software packages. All averages are ± standard error. All comparisons of proportions were made using Fisher’s exact tests. All pairwise comparisons of means were made using Student’s *t*-tests. Nosema spore counts were log_10_-transformed prior to all analyses. The average mites per 100 bees in RUS and UNSEL from pre- and post-cold storage samples was compared using analysis of variance with queen line and sample time as factors in the general linear model. Comparisons of colony sizes pre- and post-cold storage and after almond bloom were made using repeated measures analysis of variance with queen line and sample time as factors in the general linear model. Proportional changes between pre- and post-cold storage combs of brood were compared between RUS and UNSEL using a Student’s *t*-test. Relationships between pre-cold storage combs with bees and combs of brood after cold storage (all brood and sealed brood) were evaluated for each queen line using linear regression. Linear regression was also used to test for relationships between the amount of brood reared while colonies were in cold storage and pre-cold storage colony size. Slopes and y-intercepts were compared between queen lines using *t*-tests.

Comparisons of combs of bees and brood after cold storage and almond bloom were made between RUS overwintered either in cold storage or in outdoor apiaries using analysis of variance, with the sample time and overwintering method as factors in the general linear model.

Comparisons between RUS and UNSEL pre- and post-cold storage fat body weight and protein and lipid concentrations were made using analysis of variance with queen line and sample time as factors in the general linear model. Comparisons of proportional changes in fat body metrics between RUS and UNSEL during the pre- and post-cold storage interval were made using *t*-tests. We tested for relationships between changes in fat body weight and protein and lipid concentrations before and after cold storage using linear regression. In a separate analysis using linear regression, we tested for relationships between brood reared in cold storage and proportional changes in fat body weight and protein and lipid concentrations.

## 3. Results

### 3.1. Mites per 100 Bees and Nosema Spores per Colony

UNSEL colonies averaged 0.1 ± 0.1 mites per 100 bees in September, and RUS averaged 0.01 ± 0.01 mites per 100 bees. Prior to cold storage, RUS colonies averaged 0.01 ± 0.01. Mites were not detected in UNSEL. Mites per 100 bees did not differ between sample times, queen lines, or interactions between the parameters (Table 1).

UNSEL and RUS colonies did not differ in the proportion of colonies where Nosema was detected (RUS = 0.737, UNSEL = 0.687; *p* = 0.56) or in average Nosema spores per bee (RUS = 1.2 × 10^5^ ± 1.3 × 10^5^, UNSEL: 8.2 × 10^4^ ± 1.1 × 10^4^ spores per bee, t = 0.92, d.f. = 131, *p* = 0.36).

### 3.2. Colony Sizes of UNSEL and RUS Overwintered in Cold Storage

Colony sizes (combs with bees and combs of brood) did not differ between the two RUS apiaries or among the three UNSEL apiaries, so data were combined for each queen line across apiaries. Combs with bees in September did not differ between RUS and UNSEL, but RUS had significantly more combs of brood (Figure 2). During the pre- and post-cold storage interval and after almond bloom, combs with bees were affected by the queen line, sample time, and interactions between the factors. UNSEL colonies had more combs with bees during the pre-cold storage sample intervals, but after cold storage and almond bloom, combs with bees did not differ between UNSEL and RUS. While in cold storage, UNSEL colonies lost 37.0% of their adult population and RUS lost 44.3%. The proportional change between pre- and post-cold storage colony sizes did not differ between UNSEL and RUS (t_148_ =1.72, *p* = 0.09). Colony survival post-cold storage was 80% for UNSEL and 88.8% for RUS. The percentages did not differ (Fisher’s exact test *p* = 0.12). The percentage of colonies that could be rented for pollination also did not differ between UNSEL and RUS (UNSEL = 62.5%, RUS = 67.9%, *p* = 0.51). Of the colonies rented for almond pollination, 86% of UNSEL and 89% of RUS were alive after bloom, and this did not differ between queen lines (Fisher’s exact test *p* = 0.77).

UNSEL and RUS colonies were broodless when they were put into cold storage but had open and sealed brood when they were removed from the storage buildings. The total brood production after cold storage and almond bloom was not affected by queen line but was affected by sample time. There were more combs of sealed brood after cold storage in UNSEL than RUS. Combs of open and sealed brood after almond bloom did not differ between queen lines. Combs of open and sealed brood in colonies after cold storage were significantly correlated with pre-cold storage colony size (Table 2). Comparisons of regression lines derived from UNSEL and RUS colony and sealed brood measurements indicated no significant difference between y-intercepts (t = 0.66, *p* = 0.51) or slopes (t = 0.61, *p* = 0.55).

### 3.3. Fat Body Metrics

RUS colonies from the two apiary sites did not differ in fat body weight or protein or lipid concentrations, so data from the sites were pooled. Similar results occurred for the three UNSEL sites, so data were pooled. Fat body weights before and after cold storage were similar between the queen lines. Fat bodies gained weight while in cold storage and differed between pre- and post-cold storage samples (Figure 3). UNSEL gained more weight than RUS (t_48_ = 3.66, *p* = 0.001). Fat body lipid concentrations differed between queen lines and sample times. Pre-cold storage RUS lipid concentrations were significantly higher than those of UNSEL, but concentrations were similar post-cold storage. RUS lipid concentrations decreased significantly more than those of UNSEL while hives were in cold storage (t_53_ = 4.39, *p* < 0.0001). Differences between pre- and post-cold storage fat body lipid concentrations were inversely related to the amount of brood reared in overwintering RUS and UNSEL colonies (Table 3). Fat body protein concentrations also differed between queen lines and sample times. RUS had higher protein concentrations than UNSEL before cold storage and lower concentrations after. Changes in fat body weight for RUS and UNSEL were related to the differences between pre- and post-cold storage protein and lipid concentrations (RUS—weight and protein concentration: coefficient = 0.17, *p* = 0.006, weight and lipid concentration: coefficient = −3.1, *p* < 0.0001; UNSEL—weight and protein concentration: coefficient = 0.18, *p* < 0.0001, weight and lipid concentration: coefficient = −1.7, *p* < 0.005).

### 3.4. Effects of Overwintering Methods

Post-overwintering RUS and RUSms combs with bees were not affected by the overwintering site but were affected by the sampling time and the interaction between the factors (Table 4). Colonies were similar in size before and after overwintering, but RUSms averaged about one more comb with bees after almond bloom than RUS. Combs of brood after overwintering and almond bloom were significantly affected by the overwintering site and sample time. The interactions of the factors were also significant. RUSms had more brood after overwintering and almond bloom than RUS.

More RUS bees were alive after overwintering than RUSms bees, though the difference was not significant (RUS = 90%, RUSms = 77.5%; *p* = 0.057). The percentage of overwintered colonies that could be rented for almond pollination also did not differ significantly between RUS (68.7%) and RUSms (57.9%) (*p* = 0.154). After almond bloom, significantly more RUS bees were alive compared with RUSms bees (RUS = 81%, RUSms = 51.7%; *p* = <0.001).

### 3.5. Overwintering Costs

The cost of overwintering 80 colonies was USD $1369 less (USD $17 per colony) when hives were put into cold storage compared with apiaries in MA, USA (Table 5). Differences in costs came from shipping colonies to their overwintering sites (USD $752.20 more for RUSms) and then to California (USD $1093.60 more for RUSms). Overwintering costs for varroacide and feeding treatments in the apiary were comparable to cold storage fees.

Colonies were rented for almond pollination at a fee of USD $185/hive. Of the 80 colonies that were overwintered in cold storage, 55 hives could be rented for almond pollination, generating a gross revenue of USD $10,175 and net revenue of USD $6952.60 when overwintering costs were subtracted. Of the 80 colonies overwintered in Mississippi, 46 hives could be rented for almond pollination, generating a gross income of USD $8510 and a net income of USD $3918.60 after overwintering costs were subtracted.

## 4. Discussion

Comparisons between RUS and UNSEL indicated no differences in colony survival and size after cold storage overwintering. Similar percentages of RUS and UNSEL were large enough to rent for almond pollination (i.e., >6 combs of bees) and were alive after almond bloom. Fat body metrics prior to cold storage differed between RUS and UNSEL, with RUS having higher lipid but lower protein concentrations than UNSEL. Fat body metrics changed while colonies were in cold storage; weight and protein concentrations increased, and lipid concentrations decreased. The decrease in lipid concentration was positively correlated with the amount of brood reared while colonies were in cold storage. A cost analysis comparing expenditures for overwintering RUS colonies in either cold storage or outdoor apiaries indicated lower costs for colonies overwintered in cold storage. Though there were no statistical differences between cold storage and outdoor overwintering in the percentages of colonies that survived and could be rented for almond pollination, the cost analysis showed greater profit margins for overwintering in cold storage.

Colonies lost about four combs of bees while in cold storage, even though pre-cold storage samples indicated Varroa levels and Nosema spore numbers that were below the threshold that affects colony survival [38]. The stress of moving colonies to cold storage facilities and then to almond orchards along with the natural attrition of overwintering populations might account for some of the reductions in colony populations. Bees might have also been lost to drifting or the mortality of older bees after colonies were opened in the almond orchards. The causes for bee losses are speculative, however, since our post-cold storage measurements were taken after bees had been flying for several days. Still, estimating post-cold storage colony sizes several days after hives are placed in orchards is similar to when hive inspectors grade colonies for almond pollination. Losses of adult bees, even in colonies that have low Nosema and mite levels, underscore the importance of choosing only colonies with more than 12 combs with bees to overwinter in cold storage. If these colonies lose four or more combs with bees, there is still a good chance that they will meet the six-comb minimum for almond pollination rental.

RUS colonies averaged 12 combs of bees when put in cold storage, and UNSEL averaged 15. When colonies were removed from cold storage in January, RUS colonies were similar in size to UNSEL colonies, suggesting that fewer RUS bees were lost during overwintering. A study comparing the longevity of RUS and European bees in cages exposed to temperatures that cause clustering showed that RUS lived an average of 8 days longer than European bees [39]. The greater longevity of RUS compared with UNSEL while in a cluster could explain our results and suggests that more RUS than UNSEL colonies of comparable size when put into cold storage could be available for pollination rental. In our study, about 11% more RUS colonies were available for pollination rental than UNSEL.

There was a concern about putting RUS in cold storage that centered on the tendency of RUS to stop rearing brood when foragers are not bringing nectar into the hive [29]. Without some brood being reared in cold storage, colonies could decline for the entire 3–4 weeks that they pollinate almonds. We found that despite being confined in the hive without incoming resources, RUS reared brood while in cold storage. UNSEL had more sealed brood after cold storage than RUS, indicating that UNSEL may have begun rearing brood sooner. Still, the brood reared in cold storage emerged and replaced adult bees that were being lost to foraging. Instead of declining, RUS and UNSEL colony sizes increased slightly during almond bloom.

As in a previous study [30], measurements of fat body metrics before and after overwintering in cold storage revealed a shuttling of nutrients into the fat body (protein) and out of it, possibly into brood (lipid). The increase in protein concentration may have contributed to the weight gain of the fat body during cold storage. However, the source of protein that was transferred is not known. If bees were consuming stored pollen while in a cluster, it would account for the higher protein concentrations, but lipid concentrations also should have increased, but they declined. Another possibility is that protein might have been transferred to the fat body from other tissues such as the hypopharyngeal glands. These glands begin protein synthesis in the winter when brood rearing resumes [40]. Whether some of the protein is shuttled to the fat body and stored there requires further study. Unlike protein, lipid concentrations decreased during cold storage. Lipids may have been transferred to the hypopharyngeal glands and used in brood rearing since decreases in fat body lipid concentrations were correlated with the amount of brood reared in cold storage. Additional studies are needed to test these possibilities.

Though RUS and UNSEL showed similar trends in fat body metrics while in cold storage, prior to cold storage, RUS had higher lipid concentrations than UNSEL. The differences might be related to the time when RUS stopped rearing brood in relation to UNSEL. RUS and UNSEL colony sizes were similar in September. By October, RUS had 2.9 fewer combs of bees than UNSEL. Colony growth stops when brood rearing ends, and if flight weather continues, colony populations can decline. If brood rearing stopped sooner in RUS than UNSEL, RUS colonies would have been smaller in October but may have had more nutrients, particularly lipids, stored in the fat body. Previous research reported higher fat body lipid concentrations in workers from colonies that stopped rearing brood in the fall compared with those still rearing brood [30,40]. If RUS bees stop rearing brood before UNSEL, RUS might be able to be put into cold storage earlier, thus reducing the chances of mite migration and extended periods of foraging by workers. Additional studies need to be conducted to test this possibility.

If new management strategies to increase overwintering colony survival are to be adopted, they must be cost-effective. Under the conditions in our study, costs associated with overwintering colonies in cold storage were lower compared with standard overwintering in apiaries in locations with warm winters. Though the results are encouraging regarding reducing overwintering losses, additional information to improve management recommendations is needed. For example, geographic boundaries for the formation of winter bees and the time at which colonies can be put into cold storage need to be determined. The boundaries and timing will shift yearly depending on temperatures, which will become ever more mercurial in the fall due to climate change.

## 5. Conclusions

RUS and UNSEL colonies overwintered in cold storage had similar survival rates, colony sizes, and percentages that could be rented for almond pollination. Fat body metrics changed while colonies overwintered in cold storage. Fat body weights increased in relation to protein concentrations, which were higher after cold storage. Lipid concentrations in the fat body declined during cold storage in proportion to the amount of brood reared during overwintering. A cost analysis indicated that overwintering colonies in cold storage cost less than in outdoor apiaries. Colony survival and the percentage of hives that could be rented for pollination were similar between the overwintering strategies.

## Figures and Tables

**Figure 1 insects-16-00266-f001:**
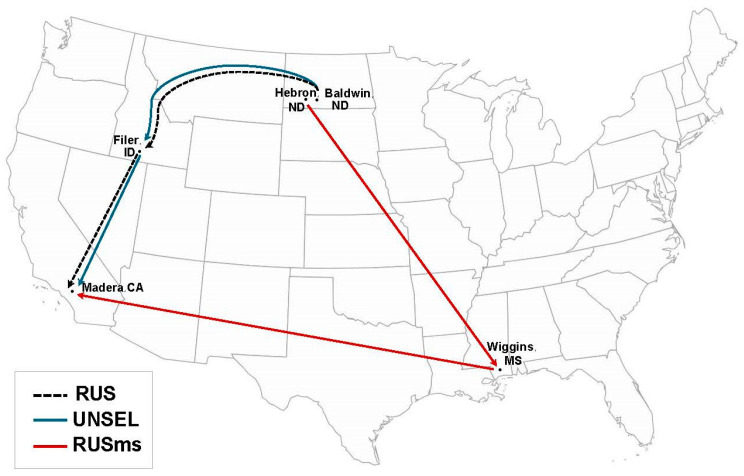
The locations of colonies used in this study. Colonies headed by unselected (UNSEL) European queens spent the summer in Baldwin, North Dakota, and those headed by Russian (RUS) queens spent the summer in Hebron, North Dakota. In October, RUS and UNSEL colonies were moved to cold storage facilities in Filer, Idaho, for overwintering. Concurrently, a group of RUS colonies were moved from Hebron, North Dakota, to outdoor apiaries in Wiggins, Mississippi, for overwintering (RUSms). In January, UNSEL, RUS, and RUSms colonies were moved to almond orchards in Madera County, California, for pollination.

**Figure 2 insects-16-00266-f002:**
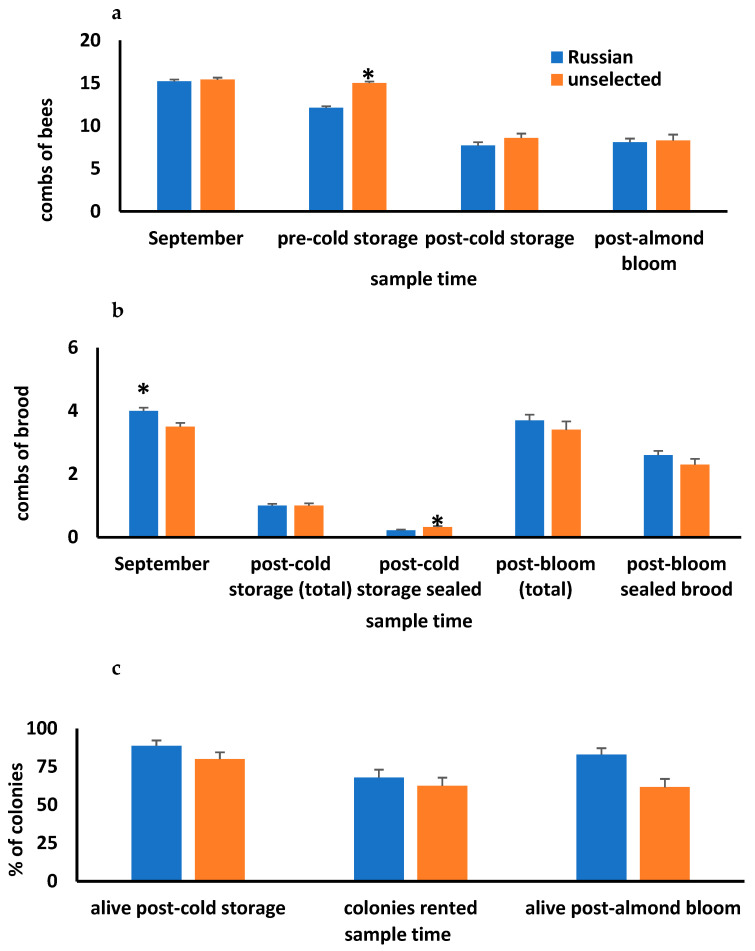
Average (with standard error bars) combs with bees (**a**) and brood (**b**) in colonies headed by either Russian (RUS) or unselected (UNSEL) European queen lines. Combs with bees were significantly affected by queen line (F_1,474_ = 9.57, *p* = 0.002), sample time (F_2,474_ = 111.4, *p* < 0.0001), and interactions between the factors (F_1,304_ = 7.19, *p* = 0.001). Combs of brood were affected by sample time (F_1,304_ = 233.8, *p* < 0.0001) but not queen line (F_1,304_ = 0.94, *p* = 0.33) or interactions between the factors (F_1,304_ = 0.82, *p* = 0.36). The average number of combs of sealed brood after cold storage was significantly higher in UNSEL than RUS (t_143_ =2.86, *p* = 0.005) but not after almond bloom (t_133_ = 1.65, *p* = 0.10). There were no differences between the percentage of RUS and UNSEL colonies that survived cold storage overwintering (*p* = 0.12), could be rented for almond pollination (*p* = 0.51), and were alive after almond bloom (*p* = 0.77) (**c**). Standard error bars are shown with the percentages. An asterisk (*) over a bar indicates significant differences at the *p* = 0.05 level between RUS and UNSEL as determined by Student’s *t*-tests.

**Figure 3 insects-16-00266-f003:**
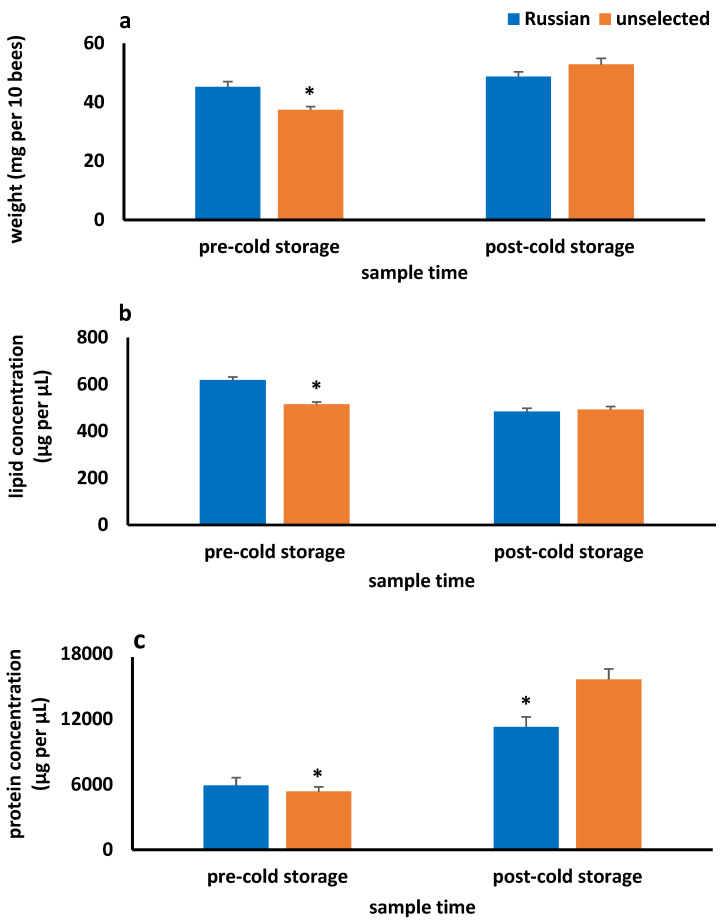
Average dry weight (**a**) and lipid (**b**) and protein (**c**) concentrations (with standard error bars) in fat bodies of worker bees from colonies of Russian and unselected European queen lines prior to (pre-cold storage) and after (post-cold storage) overwintering in cold storage. All measurements were made from pooled samples of 10 bees per colony. Fat body metrics were estimated using 30 colonies per queen line. Fat body weights did not differ between queen lines (F_1,113_ = 1.5; *p* = 0.22) but did differ between sample times (F_1,113_ = 29.0, *p* < 0.0001). The interaction term (sample time × queen line) was significant (F_1,113_ = 11.08, *p* = 0.001). Lipid concentrations differed by queen line (F_1,113_ = 13.2, *p* < 0.0001) and sample time (F_1,113_ = 36.05, *p* < 0.0001). Interactions terms were significant (F_1,113_ = 19.06, *p* < 0.0001). Protein concentrations differed by queen line (F_1,113_ = 5.40, *p* = 0.022) and sample time (F_1,113_ = 88.25, *p* < 0.0001). Interaction terms were significant (F_1,113_ = 8.66, *p* = 0.004). An asterisk (*) over a bar indicates significant differences at the *p* = 0.05 level between RUS and UNSEL as determined by Student’s *t*-tests.

**Table 1 insects-16-00266-t001:** Average Varroa mites per 100 bees ± standard error (SE) in honey bee colonies headed by either Russian or unselected European queen lines. Comparisons between queen lines were made using analysis of variance with queen line and sample time as factors in the general linear model.

Queen Line	Sample Time	Mites per 100 Bees ± SE	Factor	F	d.f	*p*
Russian	September	0.01± 0.01	Sample time	1.01	1	0.31
	Pre-cold storage (October)	0.01± 0.01	Queen line	0.52	1	0.47
Unselected	September	0.1 ± 0.1	Sample time × queen line	0.94	1	0.33
	Pre-cold storage (October)	0.00	Error		316	

**Table 2 insects-16-00266-t002:** Correlations between pre-cold storage combs of adult bees and post-cold storage combs of brood (open and sealed) in Russian and unselected European colonies overwintered in cold storage facilities.

Queen Line	Response	Predictors	Coefficient	Standard Error	d.f.	*p*	R^2^
Russian	Post-cold storage combs of brood	Pre-cold storage combs with adult bees	0.084	0.004	1, 79	<0.0001	83.0
	Post-cold storage combs of sealed brood		0.019	0.001	1, 79	<0.0001	63.8
Unselected	Post-cold storage combs of brood	Pre-cold storage combs with adult bees	0.065	0.004	1, 77	<0.0001	73.5
	Post-cold storage combs of sealed brood		0.021	0.002	1, 77	<0.0001	65.5

**Table 3 insects-16-00266-t003:** Relationships between combs of brood reared in honey bee colonies while overwintering in cold storage and changes in fat body weights and protein and lipid concentrations while colonies were overwintering. Measurements were made in colonies headed by either Russian or unselected European queen lines.

Queen Line	Fat Body Metric	Coefficient	SE of Coefficient	*p*	d.f.	R^2^
Russian	weight	0.36	0.99	0.72	1	
	protein	0.05	0.13	0.69	1	
	lipid	−3.27	0.003	0.003	1	
	regression		<0.0001	<0.0001	3	
	error				21	55.4
Unselected	weight	0.56	0.50	0.28	1	
	protein	0.19	0.10	0.07	1	
	lipid	2.09	0.91	0.03	1	
	regression				3	
	error				26	47.9

**Table 4 insects-16-00266-t004:** Comparisons of colony sizes (combs with bees and brood ± standard error) between colonies of Russian bees overwintered either in cold storage or in outdoor apiaries in Mississippi. Combs of bees prior to overwintering did not differ. Colonies did not have combs of brood during the pre-overwintering sample interval.

Parameter	Sample Time	Overwintering Site		Factor	F	d.f.	*p*
		Cold Storage	Apiary				
combs of bees	pre-overwintering	11.2 ± 0.3	10.8 ± 0.2	overwintering site	1.00	1	0.32
	post-overwintering	7.7 ± 0.4	7.5 ± 0.4	sample time	55.1	2	<0.0001
	post-almond bloom	8.1 ± 0.4	9.6 ± 0.4	overwintering site × sample time	4.08	1	0.02
				error		447	
combs of brood	post-overwintering	1.0 ± 0.06	1.8 ± 0.08	overwintering site	77.5	1	<0.0001
	post-almond bloom	3.7 ± 0.2	5.4 ± 0.2	sample time	515.5	1	<0.0001
				overwintering site × sample time	10.7	1	<0.0001
				error		280	

**Table 5 insects-16-00266-t005:** Costs (USD) incurred from overwintering colonies in either cold storage facilities or outdoors in apiaries. Colonies originated from Hebron, ND, USA, and were moved to either cold storage facilities in Filer, Idaho, or apiaries in Wiggins, MS, USA. Eighty colonies were shipped to each overwintering site. In late January, all colonies were moved to almond orchards in Madera County, CA, USA.

Service	Costs for 80 Colonies (USD) by Overwintering Method	
	Cold Storage (Cost per Colony)	Apiaries (Cost per Colony)
Shipping to overwintering site	776.80 (9.71)	1529 (19.00)
Cold storage rental	1200 (15.00)	na
Varroacide treatment in apiary	na	400 (5.00)
3 sugar syrup feedings for apiary colonies	na	640 (8.00)
1 feeding of pollen substitute for apiary colonies	na	160 (2.00)
Shipping hives: Idaho to California or Mississippi to California	768.80 (9.61)	1862.40 (23.28)
Post-cold storage sugar syrup and pollen substitute feeding	476.80 (5.96)	na
Total cost	3222.40 (40.28)	4591.40 (57.39)

## Data Availability

The original data presented in the study are openly available in AgCommons at 10.15482/USDA.ADC/28184783.

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
