# Peer review of "Adapting Overwintering Honey Bee (*Apis mellifera* L.) Colony Management in Response to Warmer Fall Temperatures Associated with Climate Change"

_insects, 2025, doi:10.3390/insects16030266_

Round 1

Reviewer 1 Report

Comments and Suggestions for Authors

Tis is a very interesting and valuable study, as it provides a solid fundation for an activity already being implemented by some beekeepers.

The objectives are clear; however, there are some detais in the descrption of materials and methods section where greater precision could enhace the study´s comprehensibility for a boarder audience.

Figure 1 requires clearer details. In this current presentation it appearse somewhat crude. It is unnecessary to include the other territories of the country, as the experiment was conduced on the mainlad. Likewise, labeling each state is not essential. Instead, the focus should be on accurately indicating locations of the apiares and the refrigerated facilities where hives were stored. It can be achieved using more aesthetically pleasing lines that do not overlap with each other.

Furthermore, it would be helpful for readers to have more details about the enviromental conditions as the sites where the hives were located before their transfer to the refrigerated facilities, as well as those in the apiary where the non-refrigerated hives were placed. Specifically, information about temperatures, type of nectar and pollen resources potentially aviable to the non refrigerated group, and if whether foraging activity was obsered during this period would provide valuble context.

Similarly, it is essential to describe the enviromental conditions under which the hives were maintained in the refrigerated facilities. At what temperature were the hives kept? Where the entrances left open? Were humidity conditions also regulated? Were there any additional enviromental controls in place, such as CO2 levels or light exposure?

In the discussion section, the content prsesented in fhe first four lines (412-415) appears unnecessary, as it repeats information already mentioned in the objectives and methodology sections.

It is mentioneted that bee losses might have been due to drifting or the mortality of older bees after colonies were opened in the almond orchards (lines 435-436). However, this observation seems somewhat speculative, as the data were collected several days after the bees had been released. While it is understandable that conducting an evaluation immediately after release would have been challeging, such timing would have provided more precise insights into the causes of bee loss. This consideration becomes particularly relevant, to distinguishing whether the observed bee losses were due to the additional handling involved in refigaration or were simply a natrural consequence of colony relocation could provide valuable insights for interpreting the results more accurately.

Author Response

Reviewer-1

Tis is a very interesting and valuable study, as it provides a solid fundation for an activity already being implemented by some beekeepers.

The objectives are clear; however, there are some detais in the descrption of materials and methods section where greater precision could enhace the study´s comprehensibility for a boarder audience.

Figure 1 requires clearer details. In this current presentation it appearse somewhat crude. It is unnecessary to include the other territories of the country, as the experiment was conduced on the mainlad. Likewise, labeling each state is not essential. Instead, the focus should be on accurately indicating locations of the apiares and the refrigerated facilities where hives were stored. It can be achieved using more aesthetically pleasing lines that do not overlap with each other.

RESPONSE: Figure 1 was changed according to the reviewer’s suggestions.

Furthermore, it would be helpful for readers to have more details about the enviromental conditions as the sites where the hives were located before their transfer to the refrigerated facilities , as well as those in the apiary where the non-refrigerated hives were placed. Specifically, information about temperatures, type of nectar and pollen resources potentially aviable to the non refrigerated group, and if whether foraging activity was obsered during this period would provide valuble context.

RESPONSE: This information was added to section 2.1 Overview of the Study immediately after each location was identified; ‘Baldwin is in central North Dakota and has a Continental climate with average temperatures between 8oC (average low) and 22oC (average high) at the start of the study in September and 0.5oC and 13oC in October. Russian colonies were sourced from two apiaries in Hebron, ND (46.867079, -101.937705). Hebron is 125km southwest of Baldwin and has a similar climate. Average September temperatures are between 6o and 22oC, and October between 13o and -1oC.’ and ‘Wiggins is located 59.5 km from the Gulf of Mexico and has a sub-tropical climate. Average low and high temperatures in November are 7o and 22oC, in December 4o and 18oC and January 2o and 16oC.’

Information on nectar and pollen sources during October-January also was added: ‘Forage available to bees included Solidago spp., Aster spp., and Baccharis halmifolia in October and November, and Alnus serrulata, Acer rubrum, Lamium spp. and Ulmus spp. in January.’

Similarly, it is essential to describe the enviromental conditions under which the hives were maintained in the refrigerated facilities. At what temperature were the hives kept? Where the entrances left open? Were humidity conditions also regulated? Were there any additional enviromental controls in place, such as CO2 levels or light exposure?

RESPONSE: Additional information about the cold storage facility was added; ‘While in the facilities, colonies were kept in darkness at an average temperature of 5.6oC maintained by refrigeration. Air exchange between the facility and the outdoors was used to keep CO2 levels to <10,000ppm, and relative humidity <50% due to the dry air in Idaho.’

In the discussion section, the content prsesented in fhe first four lines (412-415) appears unnecessary, as it repeats information already mentioned in the objectives and methodology sections.

RESPONSE: The first 4 lines of the Discussion were deleted

It is mentioneted that bee losses might have been due to drifting or the mortality of older bees after colonies were opened in the almond orchards (lines 435-436). However, this observation seems somewhat speculative, as the data were collected several days after the bees had been released. While it is understandable that conducting an evaluation immediately after release would have been challenging, such timing would have provided more precise insights into the causes of bee loss. This consideration becomes particularly relevant, to distinguishing whether the observed bee losses were due to the additional handling involved in refigaration or were simply a natrural consequence of colony relocation could provide valuable insights for interpreting the results more accurately.

RESPONSE: We agree with the reviewer that possible causes for bee losses stated in the Discussion are speculations. We modified the sentence to; The causes for bee losses are speculative however, since our post-cold storage measurements were taken after bees had been flying for several days.’ Opening colonies as soon as they are put in orchards has its own set of problems with bees leaving the hives in large numbers for cleansing flights. Allowing colonies to settle for a few days before opening them provides an estimate of colony sizes that is more in line with what hive inspectors will use to grade colonies for pollination. This information was added to the Discussion ; ‘Still, estimating post-cold storage colony sizes several days after hives are placed in orchards is similar to when hive inspectors grade colonies for almond pollination.’

Reviewer 2 Report

Comments and Suggestions for Authors

Line 123: I assume '299 RUS' refers to 299 RUS colonies. Since the authors use 'n=158' in the next line, please standardize the expression accordingly.

Line 125: So, 229 RUS and 158 UNSEL colonies had their colony size and mite level measured but only 80+80 and 80 from each group were selected for this experiment. From an experimental design perspective, it seems unusual to evaluate more colonies than needed for the sample size. Is there a reason? Was this because only around 80 colonies met the colony requirements  mentioned in line128? Otherwise, specifying the 299 and 158 colonies seems unnecessary.

Line  142: it was very confusing authors started using “queen line” here because this “term” was not explained/mentioned in the previous paragraph.

Line 144: Question here is why 62 UNSEL colonies were tested for Nosema. What is the reason behind the authors’ sampling strategy? I am asking because I see different sub-sample numbers were used for different tests.

Line 146: Were the same colonies resampled, or another set of samples were tested?

Line 180: the mite treatment info conflicts with line 141.

Line 329: Please specify if the error bars were SE or SD in this and the following figures.

Line 338: I am surprised to see the p value of post-almond bloom was the biggest but according to the bar graph the difference is the biggest. I would suggest writers double check their data.

Line 363: Authors should put tiles on the y-axis  of the figure not just units.

Line 396: why table 4 is not presented in a figure as Fig.2 and 3?

--------

This manuscript lacks a fundamental experimental design that aligns with its title and stated objectives. In addition to the concerns outlined above, several critical issues further undermine the study's validity. For example:

1.      This study aims to explore an alternative wintering strategy; however, the introduction lacks sufficient detail on how cold storage differs from traditional overwintering practices in a standard apiary.

2.      The sample size for each queen line and site is unclear. The number of colonies from each site for each queen line is not specified. Additionally, the sampling method used to select subsamples(different sample size) for different tests is not described as well. If the same colonies were followed for each test is important because eg. fat condition of colonies varies a lot even in the same site.

3.      The experimental variables were not scientifically controlled. Initially, the experiment included RUS, UNSEL, and RUSms groups. However, most of the results focused on comparisons between RUS and UNSEL, which is the approach to evaluate the effects of different queen lines and overwintering strategies on overwintering success. Nevertheless, RUSms results were only compared with RUS in section 3.4. Why were RUSms not included in other tests? Understanding the overall colony condition in RUSms would provide valuable insights.

Additionally, the overwintering costs were only compared  between RUSms and RUS raises confusion about the study's objective, leaving unclear what specific conclusions were intended.

4.      The study states that it “tested management strategy…” (line 414), which raises questions about the significance of introducing Russian bees into the research. The study did not demonstrate how the mite-resistant traits of Russian bees contributed to winter survival. Since UNSEL colonies were treated for mites, it is unsurprising that the data show no significant differences between the two queen lines. Furthermore, the mite levels were only tested at two time points: pre-selection and pre-cold storage, which were just one month apart. This limited timeframe renders these comparisons less meaningful. Testing mite levels post-storage and post-bloom would have been more relevant and valuable to the study. Overall, the mite level analysis (results 3.1) did not contribute meaningful data and appears unrelated to the study's stated objectives.

5.      The Nosema testing encountered a similar issue, as it was only conducted at the start of the study when the colonies were moved, with no follow-up evaluations. This data seems more suitable as background information for the study colonies rather than contributing to any of the conclusions.

6.      The cost analysis should include the cost per hive, as Russian colonies are known to differ in price compared to regular western colonies. Additionally, the loss of overwintered colonies should be factored into the analysis.

Author Response

Line 123: I assume '299 RUS' refers to 299 RUS colonies. Since the authors use 'n=158' in the next line, please standardize the expression accordingly.

RESPONSE: This change was made: On August 31- September 1, RUS from two apiary sites (n = 299) were measured for colony size (combs of bees and brood) and mites per 100 bees. UNSEL (n = 158 colonies) were measured and sampled for mites from September 5-7.

Line 125: So, 229 RUS and 158 UNSEL colonies had their colony size and mite level measured but only 80+80 and 80 from each group were selected for this experiment. From an experimental design perspective, it seems unusual to evaluate more colonies than needed for the sample size. Is there a reason? Was this because only around 80 colonies met the colony requirements  mentioned in line128? Otherwise, specifying the 299 and 158 colonies seems unnecessary.

RESPONSE: Yes, we evaluated more colonies than we used so we could select those with low mite numbers. We changed the sentence to: Of these, 80 colonies of either UNSEL or RUS were chosen for overwintering in cold storage because they fit our criteria of > 10 combs with adult bees, and < 1 mite per 100 bees. Smaller colonies or those with higher mite numbers were not used in the study as these factors affect colony size and survival during and after cold storage [11, 27]. A second group of 80 colonies headed by RUS that fit the colony size and mites per 100 bee criteria were selected to overwinter outdoors in apiaries in Wiggins, MS, USA (30.726688, -88.991526) (RUSms) (Figure 1).

Line  142: it was very confusing authors started using “queen line” here because this “term” was not explained/mentioned in the previous paragraph.

RESPONSE: ‘Queen lines’ was defined in section 1.1: ‘Colonies were headed by either queens of European genetic lines (i.e., queen lines) open-mated from Integri Bees/ Harvest Honey, INC., INC., Danbury TX, USA 29.212312, -95.343355) or Russian queen lines from a certified Russian queen breeder (Coy Bee Company, Wiggins MS, USA).

Line 144: Question here is why 62 UNSEL colonies were tested for Nosema. What is the reason behind the authors’ sampling strategy? I am asking because I see different sub-sample numbers were used for different tests.

RESPONSE: The n = 62 was a mistake and I appreciate the reviewer drawing my attention to it. 80 colonies of UNSEL and RUS were sampled for Nosema during the pre-cold storage sampling interval. Due to the time and expense of analyzing the fat body samples, a subsample of 30 colonies per queen line were taken. This information was added to section 1.1: Due to the time and expense of analyzing the fat body samples, a subsample of 30 colonies per queen line were taken. A second set of bee samples were taken from RUS and UNSEL (n = 80 colonies of each) to estimate Nosema spp. spores.

Line 146: Were the same colonies resampled, or another set of samples were tested?

RESPONSE: The same colonies were sampled for fat body analysis before and after cold storage. This information was added: Colony measurements and sampling for fat body analysis were repeated using the same colonies as the pre-cold storage samples after hives were removed from cold storage and placed in almond orchards for pollination during the last week in January (post-cold storage samples).

Line 180: the mite treatment info conflicts with line 141.

RESPONSE: This was an error, and I appreciate the reviewer drawing my attention to it. The corrected sentence in this section is: UNSEL colonies had been treated with amitraz formulations by the beekeeper throughout the summer and late September. RUS colonies were treated with amitraz formulations only in late September.  

Line 329: Please specify if the error bars were SE or SD in this and the following figures.

RESPONSE: In line-3 of section 2.6. Statistical Analysis, I inserted, All averages are  standard error. In Figures 1,2 and 3 captions and Table 4, it was stated that the averages were shown with standard error bars.

Line 338: I am surprised to see the p value of post-almond bloom was the biggest but according to the bar graph the difference is the biggest. I would suggest writers double check their data.

RESPONSE: The data were double checked and are correct as presented. The p-value is comparatively larger for protein concentration before and after cold storage partially due to the variation among samples within each group. The differences in variation among the fat body metrics can be seen by the size of the standard error bars. They are largest for protein concentration

Line 363: Authors should put tiles on the y-axis  of the figure not just units.

RESPONSE: Great idea! Titles were inserted followed by the units.

Line 396: why table 4 is not presented in a figure as Fig.2 and 3?

RESPONSE: I showed these data as a Table to save space rather than inserting another Figure. If the Reviewer and/or Editor would like me to make Table 4 into a Figure, I would be happy to do that.

--------

This manuscript lacks a fundamental experimental design that aligns with its title and stated objectives. In addition to the concerns outlined above, several critical issues further undermine the study's validity. For example:

  1. This study aims to explore an alternative wintering strategy; however, the introduction lacks sufficient detail on how cold storage differs from traditional overwintering practices in a standard apiary.

RESPONSE: More information was added contrasting overwintering in outdoor apiaries vs. cold storage. ‘Cold storage differs from overwintering in outdoor apiaries in that hives are moved into refrigerated buildings that are kept dark and maintained at < 7.2oC. Bees remain in cluster inside the hive while in cold storage. In contrast, colonies overwintered outdoors can experience variable temperatures and fly when weather conditions are suitable. From late fall to early spring, supplemental feeding may be needed if there is no forage available. Mite numbers may rise due to mite migration if temperatures are suitable for flight and varroacide applications might be required.’

  1. The sample size for each queen line and site is unclear. The number of colonies from each site for each queen line is not specified.

RESPONSE: The paragraphs in section 1.1 were separated into the first paragraph describing the study sites, and the second paragraph stating sample sizes for the queen lines and locations. The second paragraph states. ‘On August 31- September 1, RUS from two apiary sites (n = 299) were measured for colony size (combs of bees and brood) and mites per 100 bees. UNSEL (n = 158 colonies) were measured and sampled for mites from September 5-7. Of these, 80 colonies of either UNSEL or RUS were chosen for overwintering in cold storage because they fit our criteria of > 10 combs with adult bees, and < 1 mite per 100 bees. A second group of 80 colonies headed by RUS that fit the colony size and mites per 100 bee criteria were selected to overwinter outdoors in apiaries in Wiggins, MS…’

Additionally, the sampling method used to select subsamples(different sample size) for different tests is not described as well. If the same colonies were followed for each test is important because eg. fat condition of colonies varies a lot even in the same site.

RESPONSE: The reasons for selecting the colonies in the subsamples were described: The 80 colonies in each treatment group, ‘were chosen for overwintering in cold storage because they fit our criteria of > 10 combs with adult bees, and < 1 mite per 100 bees.’ (Paragraph 2 section 1.1). Due to the time and expense of analyzing the fat body samples, a subsample of 30 colonies per queen line were taken. A second set of bee samples were taken from RUS and UNSEL (n = 80 colonies of each) to estimate Nosema spp. spores. Bee samples for Nosema analyses were taken from honey frames. Colony measurements and sampling for fat body analysis were repeated using the same colonies as the pre-cold storage samples after hives were removed from cold storage and placed in almond orchards for pollination during the last week in January (post-cold storage samples).’ Paragraph 3 section 1.1  

  1. The experimental variables were not scientifically controlled. Initially, the experiment included RUS, UNSEL, and RUSms groups. However, most of the results focused on comparisons between RUS and UNSEL, which is the approach to evaluate the effects of different queen lines and overwintering strategies on overwintering success.

RESPONSE: The study had two components. The first was a comparison between UNSEL and RUS colony sizes, survival and percentages that could be rented for almond pollination when overwintered in cold storage. The experimental variables were carefully controlled as colonies chosen to be in the study had similar sizes and low mite and Nosema loads at the start of the study, the same experimental protocols were used to measure colony sizes, mite and Nosema spore numbers, and fat body metrics, and the colonies were overwintered in the same cold storage facility. The second part of the study focused on a cost comparison between overwintering RUS colonies in cold storage vs. outdoor apiaries in MS. If RUS could be overwintered in cold storage with success comparable to UNSEL, and the cost of cold storage overwintering is less than in MS apiaries, it could be more profitable to overwinter RUS in cold storage. These points were made in the last paragraph of the Introduction: In this study we explored two factors that could influence the decision to overwinter Russian bees in cold storage. The first was comparing colony survival after overwintering and almond bloom and the percentage of colonies that could be rented for almond pollination between Russian bees and European genetic lines previously used in cold storage overwintering experiments [30]. We chose to extend the study beyond overwintering and into almond pollination because more than one million colonies are moved into almond orchards in California for pollination each year. We also examined fat body metrics in unselected and Russian bees before and after cold storage as they provide evidence for the shift from summer to winter bees that is essential for successful overwintering in cold storage [30, 32, 33]. The second factor in the study was comparing costs and colony survival and size between Russian colonies overwintered in either cold storage or outdoor apiaries in the southern U.S. If Russian colonies could be successfully overwintered in cold storage at costs lower than or comparable to outdoor apiaries, it would provide beekeepers with management options that might mitigate the effects of climate change on colony losses.

Nevertheless, RUSms results were only compared with RUS in section 3.4. Why were RUSms not included in other tests? Understanding the overall colony condition in RUSms would provide valuable insights.

RESPONSE: The last paragraph in the Introduction was rewritten to define the two components of this study (see above). RUSms were not compared with UNSEL because an objective of the study was to compare the conventional overwintering method for RUS (outdoor apiaries in MS) with cold storage overwintering.

Additionally, the overwintering costs were only compared  between RUSms and RUS raises confusion about the study's objective, leaving unclear what specific conclusions were intended.

RESPONSE:  The following information was added to the last paragraph of the Introductions to more clearly define what specific conclusions might be drawn for the study; If Russian colonies could be overwintered in cold storage at costs lower than or comparable to outdoor apiaries, it would provide beekeepers with management options that might mitigate the effects of climate change on colony losses.’

  1. The study states that it “tested management strategy…” (line 414), which raises questions about the significance of introducing Russian bees into the research. The study did not demonstrate how the mite-resistant traits of Russian bees contributed to winter survival.

Since UNSEL colonies were treated for mites, it is unsurprising that the data show no significant differences between the two queen lines. Furthermore, the mite levels were only tested at two time points: pre-selection and pre-cold storage, which were just one month apart. This limited timeframe renders these comparisons less meaningful. Testing mite levels post-storage and post-bloom would have been more relevant and valuable to the study. Overall, the mite level analysis (results 3.1) did not contribute meaningful data and appears unrelated to the study's stated objectives.

RESPONSE: There is a lot to respond to here, so I will address the reviewer’s comments one point at a time:

The reviewer states, “:The study did not demonstrate how the mite-resistant traits of Russian bees contributed to winter survival.’

RESPONSE: Beekeepers use mite resistant lines to reduce costs of miticides and the chances that mite resistance to the treatment. Queen lines selected for mite resistance, might have other traits that could affect overwintering in cold storage such as a halt in brood rearing when resources are not coming into the colony (see P-6 in the Introduction). I can understand why the reviewer might assume that the reason we used mite resistant lines was to determine if they had traits that would contribute to overwintering survival. In fact, we stated that Russian bees might have traits that would limit their survival after cold storage (i.e., the inability to rear brood while in cold storage), but we found that this was not the case. To clarify why beekeepers use mite resistant lines of bees we added the following sentence to the Introduction: ‘The effects that climate change is having on Varroa control might be mitigated with genetic lines of Varroa resistant or tolerant honey bees. Beekeepers currently use Varroa tolerant or resistant bees to reduce the cost and possible Varroa resistance to miticide treatments that can be needed at frequent intervals from spring to fall.’ (P-4 L-2).

The reviewer states, “Since UNSEL colonies were treated for mites, it is unsurprising that the data show no significant differences between the two queen lines.

RESPONSE: What is significant about our mite counts is that mite levels were low in Russian colonies without the use of miticides.

The reviewer states, “Furthermore, the mite levels were only tested at two time points: pre-selection and pre-cold storage, which were just one month apart. This limited timeframe renders these comparisons less meaningful. Testing mite levels post-storage and post-bloom would have been more relevant and valuable to the study. Overall, the mite level analysis (results 3.1) did not contribute meaningful data and appears unrelated to the study's stated objectives.”

RESPONSE: Estimates of mites/100 bees were taken in September because colonies with high mite numbers prior to cold storage have reduced colony size and survival after cold storage. The information was relevant to the study because if there were differences in colony sizes or losses between Russian and unselected bees after cold storage or almond bloom, we needed to know if high Varroa or Nosema levels prior to overwintering contributed to these outcomes. This information was added to Materials and Methods Section 1.1 P-2 ,’ Of these, 80 colonies of either UNSEL or RUS were chosen for overwintering in cold storage because they fit our criteria of > 10 combs with adult bees, and < 1 mite per 100 bees. Smaller colonies or those with higher mite numbers were not used in the study as these factors affect colony size and survival during and after cold storage [27].’ The following information also was added to the paragraph, ‘Colonies were sampled for mites per 100 bees and Nosema spp. levels at the time of the measurements as these parasites can affect overwintering survival.

The Nosema testing encountered a similar issue, as it was only conducted at the start of the study when the colonies were moved, with no follow-up evaluations. This data seems more suitable as background information for the study colonies rather than contributing to any of the conclusions.

RESPONSE: As stated above (‘Colonies were sampled for mites per 100 bees and Nosema spp. levels at the time of the measurements as these parasites can affect overwintering survival), colony size and survival after cold storage can be affected by high Nosema spore numbers prior to cold storage, and this is why the data shown for Nosema levels are relevant to the study. As stated in response to Reviewer-1, we did not take post-cold storage colony measurements until hives had been open for several days. Obtaining good estimates of Nosema and Varroa mite levels after bees have been confined and then released is difficult as workers are reorienting to their colonies and are frequently drifting among hives (the orchards and surrounding areas had hundreds of colonies from the cold storage facilities that were not part of our study),

  1. The cost analysis should include the cost per hive, as Russian colonies are known to differ in price compared to regular western colonies. Additionally, the loss of overwintered colonies should be factored into the analysis.

RESPONSE: The reviewer states that ‘the cost analysis should include cost per hive as Russian colonies are known to differ in price compared to regular western colonies’. ‘The rental fee obtained for the Russian colonies was $185 / hive’. This is stated under Table 4.  The cost per colony was included for each Service in Table 5. The total cost per colony was added to Table 5. The loss of overwintered colonies was factored into the analysis using the costs of overwintering 80 colonies using each management method and subtracting the value from the number of colonies that could be rented for almond pollination; ‘Of the 80 colonies that were overwintered in cold storage, 55 hives could be rented for almond pollination generating a gross revenue of $10175 and net revenue when overwintering costs were subtracted of $6952.60. Of the 80 colonies overwintered in Mississippi, 46 hives could be rented for almond pollination generating a gross income of $8510 and a net income after overwintering costs of $3918.60.’

Reviewer 3 Report

Comments and Suggestions for Authors

The paper “the management of honey bee (Apis mellifera L.) colonies during overwintering to address the challenges posed by warmer fall temperatures linked to climate change” address a crucial aspect in beekeeping practice.

While the results from recent studies are promising in terms of reducing overwintering losses, additional research is necessary to further refine and enhance management recommendations.

The experimental evidence is consistent, as is the statistical analysis, but the paper definitely needs improvement.

In various sections of the text, citations are presented in an inconsistent manner, which can confuse readers. For example:

• "Varroa mites (Varroa destructor Anderson and Trueman) [8 ] [9]" should be corrected to "[8, 9]."

• "A second way warmer fall temperatures in temperate areas can threaten colony survival is through effects on the age structure of overwintering colony populations [14], [7]" should be revised to "[7, 14]."

This type of citation error should be carefully corrected throughout the entire text to improve clarity and readability.

Additionally, there should be clear correspondence between the "Materials and Methods" (M&M) section and the Results section. For instance, in the M&M section, Nosema is mentioned last, but it appears first in the Results section, which creates a disconnect. Organizing the sections so that they align better is crucial for logical flow.

A more concise and improved summary of the M&M section is also recommended to streamline the presentation without losing critical details.

In the "Fat Body Analysis" section, terminology is inconsistently used. The term "sample vial" is used interchangeably with "tube." For the sake of clarity, it would be beneficial to choose one term—either "sample vial" or "sample tube"—and use it consistently throughout the text.

The "Statistical Analysis" section could be simplified and summarized to avoid unnecessary complexity. Highlighting only the key statistical methods and results will make this section more accessible to readers.

In terms of figures, it's suggested to remove the distracting green background from the figure to improve visual clarity. Clear and simple visuals will help convey the data more effectively.

Lastly, the section discussing overwintering costs is very interesting and provides valuable insights.

 In my opinion, it will be suitable for publication in Insects only after major revisions.

SIMPLE SUMMARY/ABSTRACT

Line 13: Varroa-resistant Russian honey bees. Chack it through the text

INTRODUCTION

Lines 55-56: "Warmer fall temperatures can threaten colony survival in at least two ways."

Lines 68-69: "A second way that warmer fall temperatures in temperate areas can threaten colony survival is through effects on the age structure of overwintering colony populations [7,14]."

Lines 95-96: "Little is known about how Russian bees perform during cold storage."

Line 104: You might add a sentence to set the stage: "To compare the performance of Russian bees under different overwintering conditions, we evaluate their survival in both cold storage and outdoor apiaries."

MATERIALS AND METHODS

Line 118: the word "INC." is repeated unnecessarily.

Line 118: a braket “(“ is missing before the GPS coordinate.

Lines 126-127: "A second group of 80 RUS colonies (designated as RUSms) were selected to overwinter outdoors in apiaries in Wiggins, MS, USA (30.726688, -88.991526). (Figure 1.)"

Line 142-144: The description of the "pre-cold storage" and "post-cold storage" sampling process could be made clearer. "Bees from 30 colonies per queen line were sampled from the center combs in the hive for fat body analysis during the pre-cold storage measurements."

Lines 144-145: "Samples for Nosema spp. analyses were taken from honey frames in the hives."

Line 145: “Nosema” should not be in italics while “Nosema spp.” should.

Line 194: "The homogenizer was rinsed twice using a 500ml flask with distilled water."

Lines 196-197: "The supernatant was carefully removed from the center (clear area) of the sample in the test tube and transferred to a 1.5ml Eppendorf tube."

Lines 170-171: "From this point onward, measurements of adult populations using the California method will be referred to as ‘combs with bees’."

Fat Body Analysis section: In the Fat Body Analysis section, the term "sample vial" is used interchangeably with "tube." It would be clearer to use consistent terminology throughout. For example, either "sample vial" or "sample tube" should be used consistently.

Line 247: μg/μL

Line 223, 243, 251: replace “min” with “minutes”

262, 282: Fisher exact test (no possessive form needed)

Lines 271-273: "Proportional changes between pre- and post-cold storage combs of brood for RUS and UNSEL were compared using a Student's t-test."

Lines 277-280: "Comparisons of combs of bees and brood after cold storage and almond bloom were made between RUS overwintered either in cold storage or in outdoor apiaries using analysis of variance, with sample time and overwintering method as the two factors in the general linear model."

Lines 285-286: "Comparisons of proportional changes in fat body metrics during the pre- and post-cold storage period were made using t-tests."

Lines 287-289: "We tested for relationships between proportional changes in fat body weight, protein, and lipid concentrations, and between these metrics and combs of brood while colonies were in cold storage, using linear regression."

Statistical Analysis section: could be simplified and summarized.

RESULTS

Line 294: "RUS = 1.2 × 10^5 ± 1.3 × 10^5, UNSEL = 8.2 × 10^4 ± 1.1 × 10^4 spores per bee"

Lines 297-298: "Mites per 100 bees did not differ between sample time, queen lines, or interactions between queen lines and sample time (Table 1)."

Lines 314-316: "The percentage of overwintered colonies that could be rented for almond pollination also did not differ significantly between RUS (68.7%) and RUSms (57.9%) (p = 0.154)."

Line 316: “p” should be in italics: “p = 0.51” (check through the entire text)

Line 395: “… in addition to the cold storage fees."

Lines 400-403: "Of the 80 colonies that were overwintered in cold storage, 55 hives could be rented for almond pollination, generating a gross revenue of $10,175 and a net revenue of $6,952.60 after overwintering costs were subtracted."

Lines 403-405: "Of the 80 colonies overwintered in Mississippi, 46 hives could be rented for almond pollination, generating a gross income of $8,510 and a net income of $3,918.60 after overwintering costs."

DISCUSSION

Lines 417-419: The phrase "were large enough to rent" is not precise.

Lines 432: "The stress involved in relocating colonies to cold storage facilities and then to almond orchards..."

Line 463: please merge this phrase “Similar trends were found in our previous studies [30].” with the previous one.

Line 482: "Previous research has shown that..."

Line 494-495: Please rephrase this sentence

Author Response

The paper “the management of honey bee (Apis mellifera L.) colonies during overwintering to address the challenges posed by warmer fall temperatures linked to climate change” address a crucial aspect in beekeeping practice.

While the results from recent studies are promising in terms of reducing overwintering losses, additional research is necessary to further refine and enhance management recommendations.

The experimental evidence is consistent, as is the statistical analysis, but the paper definitely needs improvement.

In various sections of the text, citations are presented in an inconsistent manner, which can confuse readers. For example:

  • "Varroa mites (Varroa destructor Anderson and Trueman) [8 ] [9]" should be corrected to "[8, 9]."

RESPONSE: These errors were corrected throughout the manuscript.

  • "A second way warmer fall temperatures in temperate areas can threaten colony survival is through effects on the age structure of overwintering colony populations [14], [7]" should be revised to "[7, 14]."

This type of citation error should be carefully corrected throughout the entire text to improve clarity and readability.

RESPONSE: The citation was corrected as were others throughout the manuscript. 

Additionally, there should be clear correspondence between the "Materials and Methods" (M&M) section and the Results section. For instance, in the M&M section, Nosema is mentioned last, but it appears first in the Results section, which creates a disconnect. Organizing the sections so that they align better is crucial for logical flow.

RESPONSE: The mites per 100 bees and Nosema spore levels were corrected so that section 3.1 begins with mites per 100 bees followed by Nosema spore levels.

A more concise and improved summary of the M&M section is also recommended to streamline the presentation without losing critical details.

In the "Fat Body Analysis" section, terminology is inconsistently used. The term "sample vial" is used interchangeably with "tube." For the sake of clarity, it would be beneficial to choose one term—either "sample vial" or "sample tube"—and use it consistently throughout the text.

RESPONSE: The correction was made so that ‘sample vial’ was used throughout the Materials and Methods.

The "Statistical Analysis" section could be simplified and summarized to avoid unnecessary complexity. Highlighting only the key statistical methods and results will make this section more accessible to readers.

RESPONSE: The Statistical Analysis section was shortened by adding, ‘All comparisons of proportions were made using Fisher’s exact test. All pairwise comparisons of means were made using Students t-tests.’ at the beginning of the section. Specific comparisons described in the original manuscript were removed.

In terms of figures, it's suggested to remove the distracting green background from the figure to improve visual clarity. Clear and simple visuals will help convey the data more effectively.

RESPONSE: Figure 1 was replaced with a new simpler Figure.

Lastly, the section discussing overwintering costs is very interesting and provides valuable insights.

RESPONSE: Thank you.

 In my opinion, it will be suitable for publication in Insects only after major revisions.

SIMPLE SUMMARY/ABSTRACT

Line 13: Varroa-resistant Russian honey bees. Chack it through the text

RESPONSE: The correction to Varroa-resistant was changed throughout the manuscript.

INTRODUCTION

Lines 55-56: "Warmer fall temperatures can threaten colony survival in at least two ways."

RESPONSE: Change was made.

Lines 68-69: "A second way that warmer fall temperatures in temperate areas can threaten colony survival is through effects on the age structure of overwintering colony populations [7,14]."

RESPONSE: Correction was made.

Lines 95-96: "Little is known about how Russian bees perform during cold storage."

RESPONSE: Change was made.

Line 104: You might add a sentence to set the stage: "To compare the performance of Russian bees under different overwintering conditions, we evaluate their survival in both cold storage and outdoor apiaries."

 RESPONSE: The last paragraph of the Introduction was changed to include the suggestion of the reviewer, ‘The second part of the study was to compare costs and colony survival and size between Russian colonies overwintered in either cold storage or outdoor apiaries in the southern U.S. If Russian colonies could be successfully overwintered in cold storage at costs lower than or comparable to outdoor apiaries, it would provide beekeepers with management options that might mitigate the effects of climate change on colony losses.’

MATERIALS AND METHODS

Line 118: the word "INC." is repeated unnecessarily.

RESPONSE: Correction was made.

Line 118: a braket “(“ is missing before the GPS coordinate.

RESPONSE: Correction was made.

Lines 126-127: "A second group of 80 RUS colonies (designated as RUSms) were selected to overwinter outdoors in apiaries in Wiggins, MS, USA (30.726688, -88.991526). (Figure 1.)"

RESPONSE: Correction was made.

Line 142-144: The description of the "pre-cold storage" and "post-cold storage" sampling process could be made clearer. "Bees from 30 colonies per queen line were sampled from the center combs in the hive for fat body analysis during the pre-cold storage measurements."

RESPONSE: In the paragraph under the Figure 1 title, we state that, ‘RUS and UNSEL were measured again during the first week in October prior to moving hives into a refrigerated cold storage facility for overwintering (Bee Storages of Idaho LLC., Filer, ID, USA) (pre-cold storage measurement).’ In L-12 of the paragraph under the Figure 1 title, the sampling for fat body metrics was changed to, ‘Samples of bees for analysis of fat body metrics were taken from the center combs in the hive during the pre-cold storage measurements (hereafter referred to as pre-cold storage samples).’

Lines 144-145: "Samples for Nosema spp. analyses were taken from honey frames in the hives."

Line 145: “Nosema” should not be in italics while “Nosema spp.” should.

RESPONSE: Corrections were made.

Line 194: "The homogenizer was rinsed twice using a 500ml flask with distilled water."

Lines 196-197: "The supernatant was carefully removed from the center (clear area) of the sample in the test tube and transferred to a 1.5ml Eppendorf tube."

Lines 170-171: "From this point onward, measurements of adult populations using the California method will be referred to as ‘combs with bees’."

RESPONSE: Changes were made except for ‘From this point onward’. Hereafter is less wordy.

Fat Body Analysis section: In the Fat Body Analysis section, the term "sample vial" is used interchangeably with "tube." It would be clearer to use consistent terminology throughout. For example, either "sample vial" or "sample tube" should be used consistently.

RESPONSE: The correction was made so that ‘sample vial’ was used throughout the Materials and Methods.

Line 247: μg/μL

RESPONSE: either μg/μL or µg/µl is correct. I kept µg/µl.

Line 223, 243, 251: replace “min” with “minutes”

RESPONSE: min was changed to minutes

262, 282: Fisher exact test (no possessive form needed)

RESPONSE: Fisher’s exact test is used interchangeably with Fisher exact test. I kept Fisher’s exact test in the manuscript.

Lines 271-273: "Proportional changes between pre- and post-cold storage combs of brood for RUS and UNSEL were compared using a Student's t-test."

RESPONSE: Corrections were made.

Lines 277-280: "Comparisons of combs of bees and brood after cold storage and almond bloom were made between RUS overwintered either in cold storage or in outdoor apiaries using analysis of variance, with sample time and overwintering method as the two factors in the general linear model."

RESPONSE: Corrections was made.

Lines 285-286: "Comparisons of proportional changes in fat body metrics during the pre- and post-cold storage period were made using t-tests."

RESPONSE: Corrections was made.

Lines 287-289: "We tested for relationships between proportional changes in fat body weight, protein, and lipid concentrations, and between these metrics and combs of brood while colonies were in cold storage, using linear regression."

RESPONSE: Corrections was made.

Statistical Analysis section: could be simplified and summarized.

 RESPONSE: see RESPONSE above

RESULTS

Line 294: "RUS = 1.2 × 10^5 ± 1.3 × 10^5, UNSEL = 8.2 × 10^4 ± 1.1 × 10^4 spores per bee"

RESPONSE: I’d rather keep the superscripts.

Lines 297-298: "Mites per 100 bees did not differ between sample time, queen lines, or interactions between queen lines and sample time (Table 1)."

RESPONSE: I’d rather keep ‘parameters’ as it is less wordy.

Lines 314-316: "The percentage of overwintered colonies that could be rented for almond pollination also did not differ significantly between RUS (68.7%) and RUSms (57.9%) (p = 0.154)."

RESPONSE: Change was made.

Line 316: “p” should be in italics: “p = 0.51” (check through the entire text)

RESPONSE: Change was made throughout the manuscript.

Line 395: “… in addition to the cold storage fees."

RESPONSE: I do not know what this is referring to.

Lines 400-403: "Of the 80 colonies that were overwintered in cold storage, 55 hives could be rented for almond pollination, generating a gross revenue of $10,175 and a net revenue of $6,952.60 after overwintering costs were subtracted."

RESPONSE: Change was made.

Lines 403-405: "Of the 80 colonies overwintered in Mississippi, 46 hives could be rented for almond pollination, generating a gross income of $8,510 and a net income of $3,918.60 after overwintering costs."

 RESPONSE: Change was made.

DISCUSSION

Lines 417-419: The phrase "were large enough to rent" is not precise.

RESPONSE: The sentence was changed to: ‘Similar percentages of RUS and UNSEL were large enough to rent for almond pollination (i.e., > 6 combs of bees) and were alive after almond bloom.’

Lines 432: "The stress involved in relocating colonies to cold storage facilities and then to almond orchards..."

RESPONSE: I think my sentence is less wordy, and choose not to change it. The sentence is already long, and I don’t want to add more words.

Line 463: please merge this phrase “Similar trends were found in our previous studies [30].” with the previous one.

RESPONSE: The sentence was changed to: ‘As in a previous study [30], measurements of fat body metrics before and after overwintering in cold storage revealed a shuttling of nutrients into the fat body (protein) and out of it possibly into brood (lipid).

Line 482: "Previous research has shown that..."

RESPONSE: The sentence was changed to, ‘Previous research reported higher fat body lipid concentrations in workers from colonies that stopped rearing brood in the fall compared with those still rearing brood [30, 40].’

Line 494-495: Please rephrase this sentence

RESPONSE: The sentence was changed to; ‘ The boundaries and timing will shift yearly depending on temperatures that will become ever more mercurial in the fall due to climate change.’

Round 2

Reviewer 2 Report

Comments and Suggestions for Authors

Thanks for authors take time explaining and making a revision. I am OK with current version of manuscript. 

Author Response

No additional comments were made.

Reviewer 3 Report

Comments and Suggestions for Authors

The authors have thoroughly addressed each revision in a comprehensive manner. Only a minor adjustment remains: please change the background color of the graphs from green to white.

In my opinion, the manuscript is now ready for publication.

Author Response

No additional comments were made.